# Mapping expanded prostate cancer index composite to EQ5D utilities to inform economic evaluations in prostate cancer: Secondary analysis of NRG/RTOG 0415

Rahul Khairnar[1☯¤], Stephanie L. Pugh[2☯], Howard M. Sandler[3‡], W. Robert Lee[4‡], Ester Villalonga Olives[1☯], C. Daniel Mullins[1☯], Francis B. Palumbo[1☯], Deborah W. Bruner[5‡], Fadia T. Shaya[1☯], Soren M. Bentzen[6☯], Amit B. Shah[7‡], Shawn C. Malone[8‡], Jeff M. Michalski[9‡], Ian S. Dayes[10‡], Samantha A. Seaward[11‡], Michele Albert[12‡], Adam D. Currey[13‡], Thomas M. Pisansky[14‡], Yuhchyau Chen[15‡], Eric M. Horwitz[16‡], Albert S. DeNittis[17‡], Felix Y. Feng[18‡], Mark V. Mishra[19☯]*

1 Department of Pharmaceutical Health Services Research, University of Maryland School of Pharmacy, Baltimore, MD, United States of America, 2 NRG Oncology Statistics and Data Management Center, Philadelphia, PA, United States of America, 3 Department of Radiation Oncology, Cedars-Sinai Medical Center, Los Angeles, CA, United States of America, 4 Department of Radiation Oncology, Duke University, Durham, NC, United States of America, 5 Department of Radiation Oncology, Emory University, Atlanta, GA, United States of America, 6 Department of Epidemiology and Public Health, University of Maryland School of Medicine, Baltimore, MD, United States of America, 7 WellSpan Health-York Cancer Center, York, PA, United States of America, 8 Ottawa Hospital and Cancer Center, Ottawa, ON, Canada, 9 Department of Radiation Oncology, Washington University, St. Louis, MO, United States of America, 10 Juravinski Cancer Center at Hamilton Health Sciences, Hamilton, ON, Canada, 11 Kaiser Permanente Northern California, Oakland, CA, United States of America, 12 Saint Anne's Hospital, Fall River, MA, United States of America, 13 Zablocki VAMC and the Medical College of Wisconsin, Milwaukee, WI, United States of America, 14 Department of Radiation Oncology, Mayo Clinic Rochester, Rochester, MN, United States of America, 15 Department of Radiation Oncology, University of Rochester, Rochester, NY, United States of America, 16 Department of Radiation Oncology, Fox Chase Cancer Center, Philadelphia, PA, United States of America, 17 Department of Radiation Oncology, Main Line Health, Philadelphia, PA, United States of America, 18 Department of Radiation Oncology, University of California San Francisco, San Francisco, CA, United States of America, 19 Department of Radiation Oncology, University of Maryland School of Medicine, Baltimore, MD, United States of America

☯ These authors contributed equally to this work.
¤ Current address: Evidence for Access, US Medical Affairs, Genentech Inc., South San Francisco, CA, United States of America
‡ These authors also contributed equally to this work.
* mmishra@umm.edu

**Data Availability Statement:** The authors follow NRG Oncology's policies for data sharing; data request can be directed to them. NRG Oncology's

## Abstract

### Purpose

The Expanded Prostate Cancer Index Composite (EPIC) is the most commonly used patient reported outcome (PRO) tool in prostate cancer (PC) clinical trials, but health utilities associated with the different health states assessed with this tool are unknown, limiting our ability to perform cost-utility analyses. This study aimed to map EPIC tool to EuroQoL-5D-3L (EQ5D) to generate EQ5D health utilities.

### Methods and materials

This is a secondary analysis of a prospective, randomized non-inferiority clinical trial, conducted between 04/2006 and 12/2009 at cancer centers across the United States, Canada,

data sharing policy is located on their website, https://www.nrgoncology.org/Resources/Ancillary-Projects-Data-Sharing-Application. This policy follows that of the NCI. Most of the data used in this study, excluding the domain subscale scores, is already available in the NCTN/NCORP Data Archive, https://nctn-data-archive.nci.nih.gov/, as it was used in Bruner et al. 2019. The complete data used in the current study will be released in the public domain six months post publication per NCI's data sharing policy. The authors have no special access to the data and followed the NRG Oncology data sharing policy to request data.

**Funding:** This study was funded in part by the National Cancer Institute grants U10CA180868, U10CA180822 and UG1CA189867, and the American Society for Radiation Oncology (ASTRO) Comparative Effectiveness Grant. The funders had no role in study design, data collection and analysis, decision to publish, or preparation of the manuscript.

**Competing interests:** Drs. Khairnar, Albert, Bentzen, Bruner, Chen, Currey, Dayes, DeNittis, Horwitz, Lee, Michalski, Mullins, Palumbo, Pisansky, Seaward, Shah, Shaya, and Villalonga have nothing to disclose. Dr. Feng reports personal fees from Janssen Oncology, Sanofi, Bayer, Celgene, and Blue Earth Diagnostics, grants from Zenith Epigenetics, and other from PFS Genomics, outside the submitted work; Dr. Malone reports personal fees from Sanofi, and honoraria from Amgen, Abbvie, Astellas, Janssen, Tersara, Astra Zeneca, Knight Therapeutics, and Bayer, outside the submitted work; Dr. Mishra reports grants from American Society of Radiation Oncology (ASTRO), during the conduct of the study and other from Varian Medical Systems, outside the submitted work; Dr. Sandler reports grants from ACR/NRG Oncology, during the conduct of the study; personal fees from Janssen, other from Radiogel, outside the submitted work; Dr. Pugh reports other from Millennium, other from Pfizer, outside the submitted work. This does not alter our adherence to PLOS ONE policies on sharing data and materials.

and Switzerland. Eligible patients included men >18 years with a known diagnosis of low-risk PC. Patient HRQoL data were collected using EPIC and health utilities were obtained using EQ5D. Data were divided into an estimation sample (n = 765, 70%) and a validation sample (n = 327, 30%). The mapping algorithms that capture the relationship between the instruments were estimated using ordinary least squares (OLS), Tobit, and two-part models. Five-fold cross-validation (in-sample) was used to compare the predictive performance of the estimated models. Final models were selected based on root mean square error (RMSE).

## Results

A total of 565 patients in the estimation sample had complete information on both EPIC and EQ5D questionnaires at baseline. Mean observed EQ5D utility was 0.90±0.13 (range: 0.28–1) with 55% of patients in full health. OLS models outperformed their counterpart Tobit and two-part models for all pre-determined model specifications. The best model fit was: "EQ5D utility = 0.248541 + 0.000748*(Urinary Function) + 0.001134*(Urinary Bother) + 0.000968* (Hormonal Function) + 0.004404*(Hormonal Bother)– 0.376487*(Zubrod) + 0.003562*(Urinary Function*Zubrod)"; RMSE was 0.10462.

## Conclusions

This is the first study to identify a comprehensive set of mapping algorithms to generate EQ5D utilities from EPIC domain/ sub-domain scores. The study results will help estimate quality-adjusted life-years in PC economic evaluations.

## Introduction

Treatment of localized prostate cancer (PC) continues to be a major focus of public health policy debate. Patients can choose from a wide range of management options, ranging from radical prostatectomy, radiation therapy, or active surveillance [1, 2]. Survival rates do not differ significantly between the different approaches, making treatment decision-making a complex and individualized process [3, 4].

Given the high global burden of PC, there have been calls for cost-effectiveness evaluations to better understand the economic implications of PC management. Cost-effectiveness analyses (CEAs) allow for the comparison of alternative treatment options in terms of incremental costs relative to quality-adjusted life-years (QALY) gained following treatment [5]. However, such evaluations are highly dependent on our ability to not only accurately model probabilities of experiencing cancer recurrence, overall survival, and treatment side effects over time, but also our ability to accurately calculate 'utility' values associated with the range of health states that can be experienced by a patient following PC treatment. Utility values are a measure of how patients view the overall quality of their life, with '0' (corresponding to death) to '1' (corresponding to perfect health) [6]. The results of previous PC CEAs have been sensitive to the utility values attached to health states captured in the trials informing them, underscoring the need for reliable and valid utilities [7, 8].

Utilities necessary for economic evaluations can be directly elicited in trials through use of a preference-based measure (PBM) [5, 9]. However, many trials do not collect a PBM, and instead include one or more patient-reported outcome measures (PROMs), which do not have

established utility values. For example, the Expanded Prostate Cancer Index Composite (EPIC), one of the most commonly used PRO tools in prostate cancer clinical trials (including a pivotal trial comparing surgery to radiation and active surveillance [10, 11], as well as an ongoing study comparing protons to photons [12]) does not have associated utility values.

Utility mapping involves development and use of a statistical model or algorithm that links the outcomes from a PROM and a PBM to generate health utility values [5, 13–15]. Although clinical trials now often incorporate health utility estimation in their design, studies conducted in the past remain part of the evidence base as comparators for the evaluation of new technologies and have not always included a PBM [16–18]. Therefore, when utility information is not collected in a study, mapping has been proposed as an alternative solution and recommended as the second-best option after direct utility estimation for economic evaluations of interventions. The objective of this study is to map EPIC to health utilities that can be applied to future PC CEAs.

## Methods

This mapping study followed methodological guidance issued by National Institute for Health and Care Excellence (NICE), and reporting standards guidance outlined in the 2015 MAPS (MApping onto Preference-based measures reporting Standards) statement and 2017 International Society for Pharmacoeconomics and Outcomes Research (ISPOR) Task Force Report [13–15, 19]. A battery of regression model specifications were tested to identify a set of mapping algorithms with and without demographic and clinical covariates.

### Data source

The data for this study came from a previously published international multicenter, open-label randomized clinical trial (RCT) of patients with low-risk PC. This trial used a non-inferiority design to determine whether the efficacy of a hypo-fractionated treatment schedule was not worse than a conventional schedule in men with low-risk PC. The results of this trial showed no significant differences in outcomes between the two treatment modalities. Bruner et al examined the HRQoL outcomes in this trial and reported no clinically significant between-arm differences in EPIC domain scores and EQ-5D index and VAS scores through 5 years following the completion of radiation [20]. This data source was chosen for our mapping study as it collects data on both HRQoL measures of interest in PC patients undergoing treatment.

The Institutional Review Board approval was sought and received from the University Of Maryland School Of Medicine and NRG Oncology.

### Sample selection

The study sample consisted of patients who had complete information on both EPIC and EQ5D at baseline. A 70% random sample was extracted from the 1,092 analyzable patients from the trial to create the estimation cohort and the remainder 30% sample was used as a validation cohort, to predict the performance of the estimated mapping algorithms. In addition to the HRQoL data, demographic characteristics and clinical covariates were also extracted.

### Outcome measures

**EuroQol-5D-3L.** The EQ5D questionnaire is a generic PBM, recommended by NICE for use in economic evaluations and asks respondents to describe their health in five dimensions (mobility, self-care, usual activities, pain/ discomfort, and anxiety/ depression), each of which can be at one of three severity levels (1: no problems/ 2: some or moderate problems/ 3:

extreme problems) [14, 15]. Two hundred forty-three combinations can be described in this way ($3^5$ combinations). Additionally, health states corresponding to unconsciousness and immediate death are also included in the valuation process [21]. The EQ-5D tariffs for our study were obtained using the US valuation of EQ-5D health states performed by Shaw et al. in a sample of 4,048 civilian noninstitutionalized English- and Spanish-speaking adults, aged 18 and older, who resided in the United States (50 states plus the District of Columbia) in 2002 [22].

**Expanded Prostate Cancer Index Composite (EPIC).** EPIC is a comprehensive instrument designed to evaluate patient function and bother after PC treatment [3]. EPIC has been validated in men with localized PC who underwent surgery, external beam radiation, or brachytherapy with or without hormonal adjuvants. EPIC is sensitive to specific HRQoL effects of these therapies and to HRQoL effects of cancer progression [3, 23]. EPIC assesses the disease-specific aspects of PC and its therapies and is comprised of four summary domains (Urinary, Bowel, Sexual and Hormonal). In addition, each Domain Summary Score has measurable Function Subscale and Bother Subscale components. Response options for each EPIC item form a Likert scale and multi-item scale scores are transformed linearly to a 0–100 scale, with higher scores representing better HRQoL [3].

## Conceptual overlap

Pearson's correlation coefficients were used to determine the degree of conceptual overlap between EPIC domain and sub-domain scores and EQ5D index score [24, 25].

## Model development

Linear regression is the most common approach to derive mapping function [13–15]. To account for the anticipated bimodal distribution of EQ5D for our study population, other functional forms were also explored [26]. Specifically, Tobit and two-part models were estimated to account for a significant proportion of patients in full health. The Tobit model assumes that the EQ5D utility data is censored at 1 and that the true value has a normal distribution whose mean is given by a linear combination of the covariates. Two-part models model the probability of being in full health using a logistic regression, and then model the remainder of the distribution using a OLS regression model [27].

For each of the functional forms, multiple model specifications were estimated (S1 Table). Separate sets of models with EPIC domains (group 1), EPIC sub-domains (group 2), EPIC domains with demographic characteristics (group 3), EPIC sub-domains with demographic characteristics (group 4), EPIC domains with demographic characteristics and clinical covariates (group 5), and finally, EPIC sub-domains with demographic characteristics and clinical covariates (group 6) were chosen to accommodate different possible combinations of variables in EPIC datasets available to researchers. Higher second and third order polynomials for domain scores, subdomain scores, and age were explored to examine non-linear relationships; interaction terms for race and Zubrod performance status were also explored. No further covariates were explored in an effort to be able to use the mapping algorithms in a wide range of datasets. Along with the full models specified in S1 Table, reduced models were also estimated using stepwise selection (forward selection; significance level of 0.25 required for entry and to remain in the model) in order to identify parsimonious models with high predictive ability.

## Assessing model performance

The 70% random sample (n = 765) was used for estimation and internal validation of the mapping algorithms. Five-fold cross-validation was employed for estimation and internal

validation [28, 29]. In 5-fold cross-validation, the data are split into 5 equal parts and the model is fitted on 4 parts with the 5th being held out for validation. The fitted model of the 4 selected parts is used to compute the predicted residual sum of squares on the 5th omitted part, and this process is repeated for each of the 5 parts. The sum of the 5 predicted residual sums of squares is obtained for each fitted model and is the estimate of the prediction error. Indices such as the absolute mean of the residuals or errors (MAE), and square root of the mean of the residual sum of squares (RMSE) are used to determine model performance. RMSE, a measure of individual prediction error, attaches relatively higher weights to large errors, making it an ideal metric when large errors are undesirable. This study used RMSE for identifying the candidate algorithms from each of the six groups of model specifications in S1 Table. Models with lower RMSE values represent higher predictive ability. A prediction model usually performs better with the data that were used in its development. Therefore, it is critical to evaluate how well the model works in other datasets. In absence of an external dataset, validation was performed by scoring the remaining 30% random sample (n = 327) using the candidate algorithms identified using the 5-fold cross validation in the 70% estimation sample.

## Results

### Descriptive statistics

The study cohort comprised of patients who consented to QOL collection and had complete baseline data on EPIC domains/subdomains as well as EQ5D dimensions. For models with EPIC domains as the primary independent variables, 565 patients in the 70% estimation sample and 232 patients in the 30% validation sample consented and had complete baseline data on EPIC domains and EQ5D. For models with EPIC sub-domains as the primary independent variables, 507 patients in the 70% estimation sample and 213 patients in the 30% validation sample consented and had complete baseline EPIC sub-domain data and EQ5D. Patient characteristics for each of these cohorts are summarized in Table 1. EQ5D distribution was highly skewed with >50% patients in full health in each cohort; distribution plots revealed a bimodal

**Table 1. Baseline characteristics of patients with complete EPIC domain and subdomain data.**

| Characteristic | Complete EPIC domain data | | Complete EPIC sub-domain data | |
|---|---|---|---|---|
| | Estimation Cohort (n = 565) | Validation Cohort (n = 232) | Estimation Cohort (n = 507) | Validation Cohort (n = 213) |
| Continuous Variables (mean ± SD) | | | | |
| Age | 66.4±7.3 | 66.2±7.7 | 66.4±7.2 | 66.2±7.8 |
| Baseline PSA | 5.6±2.1 | 5.5±2.2 | 5.5±2.1 | 5.5±2.2 |
| Categorical Variables (n (%)) | | | | |
| Baseline PSA | | | | |
| <4 | 115 (20.3) | 45 (19.4) | 104 (20.5) | 42 (19.7) |
| ≥4 | 450 (79.7) | 187 (80.6) | 403 (79.5) | 171 (80.3) |
| Race | | | | |
| White | 466 (82.5) | 179 (77.2) | 421 (83.0) | 163 (76.5) |
| Other | 99 (17.5) | 53 (22.8) | 86 (17.0) | 50 (23.5) |
| Zubrod | | | | |
| 0 | 530 (93.8) | 211 (90.9) | 477 (94.1) | 195 (91.5) |
| 1 | 35 (6.2) | 21 (9.1) | 30 (5.9) | 18 (8.5) |
| EQ5D | | | | |
| 1 | 310 (54.9) | 120 (51.7) | 284 (56.0) | 114 (53.5) |
| <1 | 255 (45.1) | 112 (48.3) | 223 (44.0) | 99 (46.5) |

**Table 2. EPIC domain and sub-domain scores and EQ5D scores at all study time-points.**

| Characteristic | Score (Mean±SD) | |
|---|---|---|
| EPIC domains | Estimation Cohort (n = 565) | Validation Cohort (n = 232) |
| Urinary | 87.5±12.1 | 86.5±12.5 |
| Bowel | 93.4±9.3 | 92.7±9.2 |
| Sexual | 49.6±26.3 | 50.4±26.6 |
| Hormonal | 91.0±11.0 | 90.5±11.8 |
| EQ5D | 0.9±0.1 | 0.9±0.1 |
| EQ5D –median (IQR) | 1 (0.83, 1) | 1 (0.82, 1) |
| EPIC sub-domains | Estimation Cohort (n = 507) | Validation Cohort (n = 213) |
| Urinary Function | 93.3±10.7 | 92.9±11.8 |
| Urinary Bother | 84.0±14.8 | 82.7±14.8 |
| Urinary Irritation | 86.8±12.6 | 85.6±12.1 |
| Urinary Incontinence | 91.6±14.0 | 91.3±14.8 |
| Bowel Function | 93.2±8.5 | 92.3±9.4 |
| Bowel Bother | 94.6±9.6 | 93.4±10.7 |
| Sexual Function | 43.7±26.9 | 45.1±27.5 |
| Sexual Bother | 64.0±32.9 | 64.9±32.4 |
| Hormonal Function | 88.7±13.6 | 88.7±13.5 |
| Hormonal Bother | 93.0±10.3 | 92.0±10.3 |
| EQ5D | 0.9±0.1 | 0.9±0.1 |
| EQ5D –median (IQR) | 1 (0.83, 1) | 1 (0.83, 1) |

IQR = Inter-Quartile Range

distribution peaking at full health and at health utility value of 0.8 (S1 Fig). Table 2 summarizes the mean EPIC domain/ sub-domain scores in the estimation cohort and validation cohort.

## Conceptual overlap

Pearson's correlations between EQ5D and EPIC domains/ sub-domains showed evidence of conceptual overlap between the two measures. In the estimation cohort for models with EPIC domains, moderate correlations were found between EQ5D utility and urinary (r = 0.38), bowel (r = 0.34) and hormonal (r = 0.55) domains of EPIC; sexual domain was weakly correlated (r = 0.18) with EQ5D utility. In the estimation cohort for models with EPIC subdomains, low to moderate correlations were found between EQ5D and urinary function (r = 0.31), urinary bother (r = 0.36), urinary irritation (r = 0.36), urinary incontinence (r = 0.27), bowel function (r = 0.30), bowel bother (r = 0.32), hormonal function (r = 0.43), hormonal bother (r = 0.53), sexual function (r = 0.17), and sexual bother (r = 0.16).

## Mapping EPIC to EQ5D utilities

OLS, Tobit, and two-part models were estimated for all the model specifications in S1 Table, resulting in 144 unique full regression models. The best performing models for each of these regression types across the six groups of independent variables are presented in Table 3.

The OLS models outperformed the other model types in all six model specification groups. The best performing full model was an OLS model with EPIC sub-domains, age, race, Zubrod performance status, and baseline PSA levels (model 6i) with an RMSE of 0.10429:

*Predicted EQ5D = 2.922434 + 0.003627\* Urinary Function + 0.004125\* Urinary Bother – 0.003625\* Urinary irritation – 0.002242\* Urinary Incontinence – 0.0000058476\* Bowel*

**Table 3. Performance of full models in internal (5-fold cross-validation) and validation sets.**

| # | Model Specifications | | EQ5D Index Scores | | | RMSE | | Overall Rank |
|---|---|---|---|---|---|---|---|---|
| | Available Data | Regression Model | Mean ± SD | Minimum | Maximum | 5-Fold Cross-Validation | Validation | |
| | Actual EQ5D Data | - | 0.90±0.13 | 0.28 | 1.00 | - | - | - |
| 1 | EPIC Domains | **OLS (1a)** | **0.90±0.08** | **0.51** | **0.99** | **0.10819** | **0.122668** | **9** |
| | | Tobit (1b) | 0.95±0.09 | 0.37 | 1.00 | 0.12476 | - | 17 |
| | | 2-Part (1a) | 0.90±0.08 | 0.55 | 0.98 | 0.11016 | - | 11 |
| 2 | EPIC Sub-Domains | **OLS (2c)** | **0.91±0.08** | **0.33** | **1.01** | **0.10450** | **0.113311** | **2** |
| | | Tobit (2b) | 0.95±0.09 | 0.34 | 1.00 | 0.12395 | - | 14 |
| | | 2-Part (2a) | 0.91±0.08 | 0.44 | 0.98 | 0.10484 | - | 4 |
| 3 | EPIC Domains, Age, Race | **OLS (3d)** | **0.90±0.08** | **0.43** | **1.01** | **0.10818** | **0.124491** | **8** |
| | | Tobit (3j) | 0.95±0.09 | 0.46 | 1.00 | 0.12447 | - | 16 |
| | | 2-Part (3a) | 0.90±0.08 | 0.54 | 0.99 | 0.11017 | - | 12 |
| 4 | EPIC Sub-Domains, Age, Race | **OLS (4j)** | **0.91±0.08** | **0.33** | **1.01** | **0.10456** | | **3** |
| | | Tobit (4g) | 0.95±0.10 | 0.27 | 1.00 | 0.12477 | - | 18 |
| | | 2-Part (4a) | 0.90±0.08 | 0.50 | 0.99 | 0.10801 | - | 6 |
| 5 | EPIC Domains, Age, Race, Zubrod, PSA | **OLS (5g)** | **0.90±0.08** | **0.35** | **0.99** | **0.10615** | **0.122175** | **5** |
| | | Tobit (5j) | 0.94±0.09 | 0.39 | 1.00 | 0.12276 | - | 13 |
| | | 2-Part (5a) | 0.90±0.08 | 0.26 | 0.99 | 0.10838 | - | 10 |
| 6 | EPIC Sub-Domains, Age, Race, Zubrod, PSA | **OLS (6i)** | **0.91±0.08** | **0.36** | **0.99** | **0.10429** | **0.110482** | **1** |
| | | Tobit (6g) | 0.95±0.10 | 0.33 | 1.00 | 0.12407 | - | 15 |
| | | 2-Part (6a) | 0.90±0.08 | 0.51 | 0.99 | 0.10814 | - | 7 |

*Function – 0.000690\*Bowel Bother + 0.000589\*Sexual Function – 0.000244\*Sexual Bother + 0.000721\*Hormonal Function + 0.004691\*Hormonal Bother – 0.126445\*Age + 0.001997\*(Age)$^2$ – 0.000010336\*(Age)$^3$ + 0.009922\*Race(other) – 0.456669\*Zubrod + 0.016593\*Urinary Function\*Zubrod + 0.008613\*Urinary Bother\*Zubrod – 0.011\*Urinary Irritation\*Zubrod – 0.011342\*Urinary Incontinence\*Zubrod + 0.000711\*Bowel Function\*Zubrod + 0.003675\*Bowel Bother\*Zubrod – 0.001631\*Sexual Function\*Zubrod + 0.00008517\*Sexual Bother\*Zubrod – 0.000201\*Hormonal Function\*Zubrod – 0.002221\*Hormonal Bother\*Zubrod + 0.000332\*PSA (≥4)*

Reduced models for all six model specification groups were estimated to identify parsimonious models with high predictive ability (Table 4). For the reduced models, only OLS

**Table 4. Performance of reduced models in internal (5-fold cross-validation) and validation sets.**

| # | Model Specifications | | EQ5D Index Scores | | | RMSE | | Overall Rank |
|---|---|---|---|---|---|---|---|---|
| | Available Data | Regression Model | Mean ± SD | Minimum | Maximum | 5-Fold Cross-Validation | Validation | |
| | Actual EQ5D Data | - | 0.90±0.13 | 0.28 | 1.0 | - | - | - |
| 1 | EPIC Domains | U H | 0.90±0.08 | 0.51 | 0.98 | 0.10810 | 0.123367 | 5 |
| 2 | EPIC Sub-Domains | UF UB HF HB | 0.90±0.07 | 0.46 | 0.98 | 0.10631 | 0.113095 | 2 |
| 3 | EPIC Domains, Age, Race | U H | 0.90±0.08 | 0.51 | 0.98 | 0.10810 | 0.123367 | 6 |
| 4 | EPIC Sub-Domains, Age, Race | UF UB HF HB | 0.90±0.07 | 0.46 | 0.98 | 0.10631 | 0.113095 | 3 |
| 5 | EPIC Domains, Age, Race, Zubrod, PSA | U H Zubrod U\*Zubrod | 0.90±0.08 | 0.40 | 0.98 | 0.10654 | 0.123662 | 4 |
| 6 | EPIC Sub-Domains, Age, Race, Zubrod, PSA | UF UB HF HB Zubrod UF\*Zubrod | 0.90±0.08 | 0.37 | 0.97 | 0.10462 | 0.114714 | 1 |

functional form was tested as OLS full models outperformed other model types. The best performing reduced model had an RMSE of 0.10462:

*Predicted EQ5D = 0.248541 + 0.000748* Urinary Function + 0.001134* Urinary Bother + 0.000968* Hormonal Function + 0.004404* Hormonal Bother – 0.376487 * Zubrod + 0.003562* Urinary Function* Zubrod*

The candidate full and reduced models for the remaining specifications are presented in S2 Table. Validation using these candidate models resulted in slightly higher RMSE values compared to 5-fold cross-validation, but the results remained consistent with the 5-fold cross-validation (Tables 3 and 4). S2 Fig presents the plot of predicted vs. observed EQ5D utilities for the best performing models in each group. The EQ5D utilities appear to be under-predicted at higher health states and over-predicted for lower health states. However, the mean predicted EQ5D utilities were very similar to the observed EQ5D utilities.

## Discussion

This study identified a set of algorithms that map EPIC, a disease-specific HRQoL instrument in PC, to EQ5D, a generic preference-based instrument, using data from a randomized clinical trial. While there is considerable variation in the methodologies of mapping studies, a majority have employed some form of direct mapping strategy [16]. This mapping study followed the guidance from NICE and ISPOR task force and explored several functional forms and specifications to find the most straightforward model with highest predictive performance [13–15].

Tobit and two-part models were tested as their assumptions were compatible with the bimodal distribution of EQ5D utilities. However, they were outperformed by their counterpart OLS models for every model specification tested. Previous mapping studies have reported similar findings, where OLS regression provided better predictive ability than theoretically more robust regression procedures [16, 30, 31]. Separate algorithms were estimated using EPIC domains or subdomains data alone, and in combination with demographic covariates only or both demographic and clinical covariates, resulting in six unique sets of model specifications. Best-performing models for each of these sets were identified, so that researchers can use a model depending on the level of data at their disposal, thus, increasing the generalizability of this mapping exercise. In addition to the full models, reduced models were also estimated to identify parsimonious models with high predictive ability. Addition of demographic variables did not improve the predictive ability of the models; however, clinical covariates, specifically Zubrod performance status, improved the predictive performance. This was observed in both full and reduced models, where addition of clinical covariates resulted in lower RMSE values. Generally, models with EPIC sub-domains exhibited better predictive performance compared to their counterpart models with EPIC domains.

There are several strengths of this study that are worth mentioning. To the best of our knowledge, this is the first study to map EPIC to obtain health utilities for patients with PC. Bremner at al. mapped Prostate Cancer Index (PCI) to Patient-Oriented Prostate Utility Scale (PORPUS-U) utilities to incorporate historically collected HRQoL data in longitudinal datasets such as CaPSURE in economic evaluations [7]. EPIC is a more comprehensive instrument that evolved from PCI and is the most widely used PC specific HRQoL instrument in trials and clinical practice [23]. The algorithms identified in this study will allow incorporation of a vast body of evidence on comparative effectiveness of PC treatments in future economic evaluations. EQ5D is the recommended PBM by HTA bodies such as NICE, and considerable differences exist, even between utilities derived from different generic PBMs. Inconsistencies in the choice of PBMs in mapping studies would make comparisons across treatments and

disease areas difficult. Unlike Bremnen et al., EQ5D, a generic PBM, was chosen in order to make comparisons across disease areas possible.

Mapping algorithms perform best when the target population has characteristics similar to the source population. While the trial sample does not represent every PC patient, a large proportion of patients with PC fall in this category. Patients with low-risk PC, as in this sample, tend to have high performance status and high EQ5D scores with minimal variability which may differ substantially from high-risk patients. Thus, caution should be exercised in extrapolating these algorithms to patients with high-risk PC. Future analyses could build on this work and identify best performing models for patients with high-risk PC.

As with any mapping study, this study has some limitations that merit discussion. Validation of candidate models in the 30% sample resulted in slightly higher RMSE values than those observed in the estimation cohort. This was expected as prediction models usually perform better with the data that were used in its development. However, models with lower RMSE values in the 5-fold cross-validation also had lower RMSE values in the validation set, supporting the robust predictive performance of the candidate algorithms in external datasets. While the health utilities for milder health states were under-predicted and worse health states were over-predicted, the mean predicted utilities at the population level were very similar to the observed mean utilities. These mapping algorithms are best suited to predict mean utilities and may not predict individual level EQ5D utilities with high degree of accuracy. Finally, inclusion in the estimation sample required complete data on EPIC domains/sub-domains along with EQ5D utilities. While differences may exist between patients who completed the questionnaires versus those who did not, the objective of our regression models was prediction and not estimation, therefore, risk of bias is minimal with using this subset of patients. There is considerable heterogeneity in the data sources that have been used in mapping studies; future studies should compare the impact of these differences on the resulting algorithms.

In conclusion, HRQoL measures can be descriptive (generic, or condition-specific) or preference-based (health utility measures) [32, 33]. It is often not feasible to include all these types of instruments in a given study, as this can be a costly and time-consuming endeavor. These studies however, form an important part of the evidence base for the effectiveness of an intervention. Mapping EPIC to EQ5D utilities bridges an important outcomes gap, allowing incorporation of a vast body of literature measuring descriptive HRQoL data in PC patients in the healthcare decision-making process.

## Supporting information

**S1 Fig. Distribution plot of EQ5D in the estimation cohorts.** A. Patients with Complete Epic Domain Data (N = 565). B. Patients with Complete Epic Sub-Domain Data (N = 507).
(ZIP)

**S2 Fig. Plot of observed vs. predicted EQ5D utilities for candidate full models.**
(ZIP)

**S3 Fig. Bland-Altman plot for full candidate model using EPIC sub-domain data.**
(TIF)

**S1 Table. Model specifications.**
(DOCX)

**S2 Table. Candidate mapping algorithms and external validation results in the 30% sample.**
(DOCX)

**S3 Table. Baseline characteristics of patients with complete EPIC domain data.**
(DOCX)

**S4 Table. EQ5D and EPIC domain scores for patients with complete EPIC domain data.**
(DOCX)

**S5 Table. EPIC sub-domain scores for patients with complete EPIC sub-domain data.**
(DOCX)

**S6 Table. Baseline characteristics of patients included vs not included in complete EPIC sub-domain analysis.**
(DOCX)

## Acknowledgments

We acknowledge Lyudmila DeMora, MS, for her statistical support with validation.

## Author Contributions

**Conceptualization:** Rahul Khairnar, Ester Villalonga Olives, C. Daniel Mullins, Francis B. Palumbo, Fadia T. Shaya, Soren M. Bentzen, Amit B. Shah, Mark V. Mishra.

**Data curation:** Rahul Khairnar, Stephanie L. Pugh, Howard M. Sandler, W. Robert Lee, Ester Villalonga Olives, C. Daniel Mullins, Francis B. Palumbo, Deborah W. Bruner, Fadia T. Shaya, Soren M. Bentzen, Amit B. Shah, Shawn C. Malone, Jeff M. Michalski, Ian S. Dayes, Samantha A. Seaward, Michele Albert, Adam D. Currey, Thomas M. Pisansky, Yuhchyau Chen, Eric M. Horwitz, Albert S. DeNittis, Felix Y. Feng, Mark V. Mishra.

**Formal analysis:** Rahul Khairnar, Stephanie L. Pugh.

**Funding acquisition:** Rahul Khairnar, Howard M. Sandler, W. Robert Lee, Ester Villalonga Olives, C. Daniel Mullins, Francis B. Palumbo, Deborah W. Bruner, Fadia T. Shaya, Soren M. Bentzen, Amit B. Shah, Shawn C. Malone, Jeff M. Michalski, Ian S. Dayes, Samantha A. Seaward, Michele Albert, Adam D. Currey, Thomas M. Pisansky, Yuhchyau Chen, Eric M. Horwitz, Albert S. DeNittis, Felix Y. Feng, Mark V. Mishra.

**Investigation:** Rahul Khairnar, Stephanie L. Pugh, Howard M. Sandler, W. Robert Lee, Ester Villalonga Olives, C. Daniel Mullins, Francis B. Palumbo, Deborah W. Bruner, Fadia T. Shaya, Soren M. Bentzen, Amit B. Shah, Shawn C. Malone, Jeff M. Michalski, Ian S. Dayes, Samantha A. Seaward, Michele Albert, Adam D. Currey, Thomas M. Pisansky, Yuhchyau Chen, Eric M. Horwitz, Albert S. DeNittis, Felix Y. Feng, Mark V. Mishra.

**Methodology:** Rahul Khairnar, Stephanie L. Pugh, Howard M. Sandler, W. Robert Lee, Ester Villalonga Olives, C. Daniel Mullins, Francis B. Palumbo, Deborah W. Bruner, Fadia T. Shaya, Soren M. Bentzen, Amit B. Shah, Shawn C. Malone, Jeff M. Michalski, Ian S. Dayes, Samantha A. Seaward, Michele Albert, Adam D. Currey, Thomas M. Pisansky, Yuhchyau Chen, Eric M. Horwitz, Albert S. DeNittis, Felix Y. Feng, Mark V. Mishra.

**Project administration:** Rahul Khairnar, Stephanie L. Pugh, Howard M. Sandler, W. Robert Lee, Ester Villalonga Olives, C. Daniel Mullins, Francis B. Palumbo, Deborah W. Bruner, Fadia T. Shaya, Soren M. Bentzen, Amit B. Shah, Shawn C. Malone, Jeff M. Michalski, Ian S. Dayes, Samantha A. Seaward, Michele Albert, Adam D. Currey, Thomas M. Pisansky, Yuhchyau Chen, Eric M. Horwitz, Albert S. DeNittis, Felix Y. Feng, Mark V. Mishra.

**Resources:** Rahul Khairnar, Stephanie L. Pugh, Howard M. Sandler, W. Robert Lee, Francis B. Palumbo, Deborah W. Bruner, Soren M. Bentzen, Amit B. Shah, Shawn C. Malone, Jeff M. Michalski, Ian S. Dayes, Samantha A. Seaward, Michele Albert, Adam D. Currey, Thomas M. Pisansky, Yuhchyau Chen, Eric M. Horwitz, Albert S. DeNittis, Felix Y. Feng, Mark V. Mishra.

**Software:** Rahul Khairnar, Samantha A. Seaward.

**Supervision:** Rahul Khairnar, Stephanie L. Pugh, Howard M. Sandler, W. Robert Lee, Ester Villalonga Olives, C. Daniel Mullins, Francis B. Palumbo, Deborah W. Bruner, Fadia T. Shaya, Soren M. Bentzen, Amit B. Shah, Shawn C. Malone, Jeff M. Michalski, Ian S. Dayes, Samantha A. Seaward, Michele Albert, Adam D. Currey, Thomas M. Pisansky, Yuhchyau Chen, Eric M. Horwitz, Albert S. DeNittis, Felix Y. Feng, Mark V. Mishra.

**Validation:** Rahul Khairnar, Stephanie L. Pugh, Howard M. Sandler, W. Robert Lee, C. Daniel Mullins, Francis B. Palumbo, Fadia T. Shaya, Soren M. Bentzen, Jeff M. Michalski, Ian S. Dayes, Samantha A. Seaward, Michele Albert, Adam D. Currey, Thomas M. Pisansky, Yuhchyau Chen, Eric M. Horwitz, Albert S. DeNittis, Felix Y. Feng, Mark V. Mishra.

**Visualization:** Rahul Khairnar, Stephanie L. Pugh, Howard M. Sandler, W. Robert Lee, Ester Villalonga Olives, C. Daniel Mullins, Francis B. Palumbo, Deborah W. Bruner, Fadia T. Shaya, Soren M. Bentzen, Amit B. Shah, Shawn C. Malone, Jeff M. Michalski, Ian S. Dayes, Samantha A. Seaward, Michele Albert, Adam D. Currey, Thomas M. Pisansky, Yuhchyau Chen, Eric M. Horwitz, Albert S. DeNittis, Felix Y. Feng, Mark V. Mishra.

**Writing – original draft:** Rahul Khairnar, Mark V. Mishra.

**Writing – review & editing:** Rahul Khairnar, Stephanie L. Pugh, Howard M. Sandler, W. Robert Lee, Ester Villalonga Olives, C. Daniel Mullins, Francis B. Palumbo, Deborah W. Bruner, Fadia T. Shaya, Soren M. Bentzen, Amit B. Shah, Shawn C. Malone, Jeff M. Michalski, Ian S. Dayes, Samantha A. Seaward, Michele Albert, Adam D. Currey, Thomas M. Pisansky, Yuhchyau Chen, Eric M. Horwitz, Albert S. DeNittis, Felix Y. Feng, Mark V. Mishra.

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
