## [Decision Letter · Decision Letter 0]

16 Dec 2020

PONE-D-20-34900

Mapping Expanded Prostate Cancer Index Composite to EQ5D Utilities to Inform Economic Evaluations in Prostate Cancer: Secondary Analysis of NRG/RTOG 0415

PLOS ONE

Dear Dr. Mishra,

Thank you for submitting your manuscript to PLOS ONE. After careful consideration, we feel that it has merit but does not fully meet PLOS ONE’s publication criteria as it currently stands. Therefore, we invite you to submit a revised version of the manuscript that addresses the points raised during the review process.

We look forward to receiving your revised manuscript.

Kind regards,

Michael E. O'Callaghan

Academic Editor

PLOS ONE

Journal Requirements:

2. Within the Methods section, please provide additional details the methodology used for the selection of the published international multicentred, open-label randomised clinical trials.

"Drs. Khairnar, Albert, Bentzen, Bruner, Chen, Currey, Dayes, DeNittis, Horwitz, Lee, Michalski, Mullins, Palumbo, Pisansky, Seaward, Shah, Shaya, and Villalonga have nothing to disclose. Dr. Feng reports personal fees from Janssen Oncology, Sanofi, Bayer, Celgene, and Blue Earth Diagnostics, grants from Zenith Epigenetics, and other from PFS Genomics, outside the submitted work; Dr. Malone reports personal fees from Sanofi, and honoraria from Amgen, Abbvie, Astellas, Janssen, Tersara, Astra Zeneca, Knight Therapeutics, and Bayer, outside the submitted work; Dr. Mishra reports grants from American Society of Radiation Oncology (ASTRO), during the conduct of the study and other from Varian Medical Systems, outside the submitted work; Dr. Sandler reports grants from ACR/NRG Oncology, during the conduct of the study; personal fees from Janssen, other from Radiogel, outside the submitted work; Dr. Pugh reports other from Millennium, other from Pfizer, outside the submitted work. "

Reviewers' comments:

Reviewer's Responses to Questions

**Comments to the Author**

1. Is the manuscript technically sound, and do the data support the conclusions?

Reviewer #1: No

Reviewer #2: Yes

2. Has the statistical analysis been performed appropriately and rigorously? 

Reviewer #1: No

Reviewer #2: Yes

3. Have the authors made all data underlying the findings in their manuscript fully available?

Reviewer #1: No

Reviewer #2: No

4. Is the manuscript presented in an intelligible fashion and written in standard English?

Reviewer #1: Yes

Reviewer #2: Yes

5. Review Comments to the Author

Reviewer #1: Mapping prostate cancer index

General comments:

* replace everywhere "external validation" by "test sample", this is the sample for internal validation.

Refer to the first sample as the "estimation sample", this is the equivalent to a training sample in data science also called the hold out sample

external validation implies the use of an independent external data sample which is not the case here

(see https://en.wikipedia.org/wiki/Training,_validation,_and_test_sets)

* replace everywhere in the text EQ5D by EQ-5D-3L as this is the version of the PBM questionnaire

METHODS

Sample Selection :

* Shortly describe the sample and the original data plus QoL results of the original article

* correct the naming of the samples (see above)

Outcome measures :

* specify which value set (Tariff) was used to value the EQ-5D-3L data

* I am not sure what you mean by "not part of the descriptive system"; death is anchored at zero and states worse than death can take negative values up to a lower limit , depending on the country specific Tariff used .

Model Development :

* line 160: this is wrong; OLS assumption implies that the errors (conditional on the explanatory variables) is normally distributed. This allows inference of the coefficients and tests of significance based on Normal Theory.

*Put the detailed list of the model specifications in an appendix

* line 173: higher second and third order polynomials

* which stepwise variable selection was used ? It seems it was a simple forward selection method based on the p-value. what was the criterion used to include/retain a variable ?

(see Forward selection is a type of stepwise regression which begins with an empty model and adds in variables one by one. In each forward step, you add the one variable that gives the single best improvement to your model.Sep 19, 2017

* line 166 the exact two-step method should be more detailed

was logistic regression used for the full health, what was its accuracy ?

how was the goodness of fit of the combined parts estimated utilities then further assessed ?

Were there any U values = 1 or higher resulting form the OLS regression part ? how were these dealth with ?

Forward Selection: Definition - Statistics How To

www.statisticshowto.com › forward-selection

Stepwise regression - Wikipedia "en.wikipedia.org › wiki › Stepwise_regression"

see Variable Selection www.biostat.jhsph.edu › ~iruczins › teaching

and for a discussion and limitations of the different methods

Loann. D. Desboulets , A Review on Variable Selection in Regression Analysis Econometrics, 2018

The results of the forward regresson methods should be confrmed by another selection method, especially since the n/p (observations/parameters) ratio is rather low in the regression incorporating interactions and power variables.

A LASSO type or similar method would be useful in this situation. It can be applied to the full most detailed equation including all subscales and interactions/power variables.

Assessing Model Performance :

The choice of only the RMSE as accuracy criterion (Goodness of fit) is not to be recommended , it should be complemented by other criteria as well including MAE, estimated utility values >1 and <0, etc...

I would urge the authors to also present a Bland-Altman plot of the results of their best fitting model (with 95% confidence intervals and minimally clinically important limits as well for EQ-5D utilities +- 0.08)

Also the multiple comparison problem given the hughe number of regressions performed should be discussed/adressed .

Five-fold cross-validation was used , I guess this was on the test sample ? how were the regression results then combined ? give some more details about the exact procedure followed to allow replication of your methodology by others.

RESULTS

Descriptive Statistics :

* Include the results of a statistical comparison test of the variables between the different samples in table 2 and table 3 to assess their similarity of the samples

* given the highly bimodal nature of the observed utilities non-parametric summary measures (medians, IQR, etc..) and test statistics should be preferred added to the tables

Mapping results :

* show first the tables of performance and selection of the best fitting equations

then show the detailed equation of the best fitting one(s)

A likelihood ratio test should be performed to compare the reduced equation and the full equation of the predicted 5EQ-5D as these are nested. If the H0 of equality is not rejected (in the testing sample) then the full equation can be dropped

* present the regression coefficients with their 95% CI and present aso the variance-covariance matrix of the regression parameters

DISCUSSION

line 316 which generic PBM did Bremen used ? specify

how bad was the underprediction of low observed utilities ? where was the utility threshold ?

how bad was the overprediction for observed high utilities ? where was the utility threshold ?

What was the variance of the estimates compaed to the observed variance of utilities for different values of utilities (low, average ,high, perfect health) or per quartile?

line 338: your risk of bias is linked to whether the censoring and non-response to the QoL questionnaires was truly random otherwise there is a risk of "survival or response" bias. Nothing tells you that the non responders had the same mapping coefficients as those of the completers so this could potentially alter the regression coefficients.

Reviewer #2: This paper has examined three econometric models for estimating EuroQol- 5 Dimension (EQ-5D) utility scores from the Expanded Prostate Cancer Index Composite (EPIC) to calculate quality adjusted life years for cost-utility analysis. The paper uses robust methods that should act as an aid for utility estimation within future economic evaluations of interventions using the Expanded Prostate Cancer Index Composite in Prostate Cancer. As such, it has the potential to act as a beneficial addition to the mapping literature. This article is well written, and the authors have carefully followed standard mapping methodology.

Major comments:

1. Abstract Page 3, Line 51: The authors state that the lack of health utilities associated with the different health states assessed with the EPIC are unknown, therefore limiting the ability to perform cost-effectiveness evaluations. Can the authors edit this and use cost-utility analysis (CUA) and not cost-effectiveness analysis (CEA) as the form of economic evaluation which allows for the comparison of alternative treatment options in terms of incremental costs relative to quality-adjusted life-years (QALY) gained following treatment is a cost-utility analysis.

2. Abstract Page 3, Line 52: The authors use the term "utility weights". This term is used in valuation studies when generating population preference weights or scoring algorithms and not mapping algorithms. The authors should correct this and use utility scores or utilities instead.

3. Page 6: The authors present mapping as though the reader might already know what it is. Can the authors provide a more detailed definition of what mapping is.

4. Page 8 Line 162: Several other estimators have been applied in the mapping literature, including Fractional Logistic regression (FLOGIT); Censored Least Absolute Deviations (CLAD) regression; Generalized Additive Models; and finite mixture models. There are critics of the Tobit estimators, for example, but why haven't finite mixture models been applied?

5. Details of ethics committee approvals should be provided.

6. Model selection should not be based solely upon the criteria, such as the predictive accuracy of on root mean square error (RMSE), laid out on page 12. The paper would be strengthened by a formal and staged selection process employed to choose between the models, including the BIC, AIC (for models for which the likelihood can be computed), misspecification tests, comparisons of conditional means or other similarly informative measures. These should dictate both the choice of covariates as well as the selection across different models.

7. Page 15, When assessing model performance: the errors should also be reported across subsets of the EQ5D utility score range as this is useful for indicating whether or not there is systematic bias in the predictions.

8. External validation is the preferred method for ascertaining the predictive accuracy of a mapping model. The authors of this paper use in-sample validation methods. Can the authors provide a detailed explanation of what a "five-fold cross-validation" is and how the in-sample validation datasets were generated? Secondly, how did they ensure that 'overfitting' was not an issue in the validation exercise? Thirdly, can the authors comment on how adequate five-fold cross-validation is as opposed to say ten-fold validation which has been in several mapping studies.

Minor comments:

1. Figure S1A: Please correctly label x-axis EQ5D and not EQ5D0

2. Page 8 Line 154: Please correct HRQOL to HRQoL

3. The paper does not seem to fully get across that mapping is a second-best solution and that having original data collected from relevant populations is a better solution. For the uninitiated, they may believe that EPIC data collected from patients with PC can be converted to EQ5D utilities "with a high level of accuracy". Hence, there is no need to collect original utility data.

6. PLOS authors have the option to publish the peer review history of their article (what does this mean?). If published, this will include your full peer review and any attached files.

Reviewer #1: No

Reviewer #2: No

---

## [Author Response · Author response to Decision Letter 0]

11 Feb 2021

Response to Reviewer’s Comments

Dear Reviewers,

We thank you for your insightful feedback. We have incorporated it where possible and provided justifications for our approach where needed. We’ve summarized the responses to your comments below.

Reviewer #1: Mapping prostate cancer index

General comments: 

1. Replace everywhere "external validation" by "test sample", this is the sample for internal validation. Refer to the first sample as the "estimation sample", this is the equivalent to a training sample in data science also called the hold out sample. External validation implies the use of an independent external data sample which is not the case here. (see https://en.wikipedia.org/wiki/Training,_validation,_and_test_sets).

Answer: We agree that we performed an added internal validation step in absence of an external dataset to perform external validation. We’ve made the suggested edit wherever it applies. We have replaced ‘external validation’ with ‘validation’ and have referred to the internal validation using the estimation cohort as ‘5-fold cross-validation’.

2. Replace everywhere in the text EQ5D by EQ-5D-3L as this is the version of the PBM questionnaire

Answer: To address this, we use EQ-5D-3L (EQ5D) the first time it is referenced and EQ5D subsequently.

METHODS

Sample Selection:

3. Shortly describe the sample and the original data plus QOL results of the original article

Answer: RTOG 0415 (Lee et al.) is a non-inferiority trial to determine whether the efficacy of a hypo-fractionated treatment schedule was not worse than a conventional schedule in men with low-risk PC. Using QOL data from this trial, Bruner et al reported no clinically significant between-arm differences in EPIC domain scores and EQ-5D index and VAS scores through 5 years following the completion of radiation. Taken together with the reporting by Lee et al, treatment with HRT is non-inferior to CRT in terms of disease free survival and prostate cancer-specific and general QOL, providing evidence to affirm that HRT is the standard of care in men with low-risk prostate cancer.

We added the following text in the manuscript to summarize the key take away from the trial: “The results of the trial showed no significant differences in outcomes between the two treatment modalities.”

4. Correct the naming of the samples (see above):

Answer: Addressed. Appropriate changes were made wherever applicable.

Outcome measures:

5. Specify which value set (Tariff) was used to value the EQ-5D-3L data

Answer: We computed the U.S. preference-weighted index score, see: Shaw JW, Johnson JA, Coons SJ. US valuation of the EQ-5D health states: development and testing of the D1 valuation model. Med Care. 2005 Mar;43(3):203-20. 

The following sentence was added to the manuscript: “The EQ-5D tariffs for our study were obtained using the US valuation of EQ-5D health states performed by Shaw et al. in a sample of 4,048 civilian noninstitutionalized English- and Spanish-speaking adults, aged 18 and older, who resided in the United States (50 states plus the District of Columbia) in 2002.”

6. I am not sure what you mean by "not part of the descriptive system"; death is anchored at zero and states worse than death can take negative values up to a lower limit, depending on the country specific Tariff used.

Answer: Addressed. We deleted “not part of the descriptive system’ from the text to avoid any confusion. 

Model Development:

7. Line 160: this is wrong; OLS assumption implies that the errors (conditional on the explanatory variables) is normally distributed. This allows inference of the coefficients and tests of significance based on Normal Theory.

Answer: Addressed. We removed the incorrect statement.

8. Put the detailed list of the model specifications in an appendix

Answer: Addressed. Moved the table of specifications from the manuscript to the appendix.

9. Line 173: higher second and third order polynomials

Answer: Addressed. Added “second and third”.

10. Which stepwise variable selection was used? It seems it was a simple forward selection method based on the p-value. What was the criterion used to include/retain a variable?

(See Forward selection is a type of stepwise regression which begins with an empty model and adds in variables one by one. In each forward step, you add the one variable that gives the single best improvement to your model.

Answer: We used the forward selection method in which we began with an empty model and added variables one after the other. The p-value for entry was set at 0.25 and the p-value for retention was kept at 0.25. We’ve added text to the manuscript to reflect this.

11. Line 166 the exact two-step method should be more detailed. Was logistic regression used for the full health, what was its accuracy? How was the goodness of fit of the combined parts estimated utilities then further assessed? Were there any U values = 1 or higher resulting from the OLS regression part? How were these dealt with?

Answer: We have added text to the manuscript to reflect that a logistic regression was conducted to identify people in full health. As several models were run, the accuracy varied across models but around 80% of the patients who were full health were correctly classified. 

The final model selection was done based on how well the utilities were predicted in both parts put together and was assessed using RMSEs and MAEs.

There were some cases where the predicted utilities were over 1; these potentially contributed to higher RMSE values for two-part models.

12. Forward Selection: Definition - Statistics How To

www.statisticshowto.com › forward-selection

Stepwise regression - Wikipedia "en.wikipedia.org › wiki › Stepwise_regression"

see Variable Selection www.biostat.jhsph.edu › ~iruczins › teaching

and for a discussion and limitations of the different methods

Loann. D. Desboulets , A Review on Variable Selection in Regression Analysis Econometrics, 2018

Answer: We thank the reviewer for sharing these useful resources.

13. The results of the forward regresson methods should be confrmed by another selection method, especially since the n/p (observations/parameters) ratio is rather low in the regression incorporating interactions and power variables.

A LASSO type or similar method would be useful in this situation. It can be applied to the full most detailed equation including all subscales and interactions/power variables.

Answer: We thank the reviewer for their insights about this method to obtain simpler models. We did not consider applying LASSO regularization to our models, as we examined models ranging from just a few parameters to a large number of parameters.

Assessing Model Performance:

14. The choice of only the RMSE as accuracy criterion (Goodness of fit) is not to be recommended, it should be complemented by other criteria as well including MAE, estimated utility values >1 and <0, etc.

Answer: While both RMSE and MAE have been used to compare predictive performance of mapping algorithms, RMSE penalizes larger errors more than MAE, making it a more appropriate metric to assess overall performance of mapping algorithms. However, we also computed the MAEs for the models and the results were mostly consistent with the RMSEs. Algorithms in this study rarely yielded predicted utilities higher than 1, and no action in that regard was needed as a result.

15. I would urge the authors to also present a Bland-Altman plot of the results of their best fitting model (with 95% confidence intervals and minimally clinically important limits as well for EQ-5D utilities +- 0.08)

Answer: Addressed. Bland-Altman plot for the best fitting model is provided in the supporting material. The green lines reflect the MCID of 0.08 for EQ5D suggested by the reviewer.

16. Also the multiple comparison problem given the huge number of regressions performed should be discussed/addressed.

Answer: In this analysis, no multiplicity corrections were taken into account since this process concerned building the most appropriate model as opposed to interpretation of the results of the model. When building and comparing different models, inflation of the type I error was irrelevant since we were not assessing the significance of any particular variable but the overall fit of the model.

17. Five-fold cross-validation was used, I guess this was on the test sample? How were the regression results then combined? Give some more details about the exact procedure followed to allow replication of your methodology by others.

Answer: For the OLS models, the PROC GLMSELECT procedure was used and 5-fold cross-validation was performed using “CV Method = block (5)” option. For the Tobit and Two-part models, SAS macros were used to split the sample, run the regressions in training sets, score the test sets, and combine the estimates. The SAS code for this analysis can be provided upon request. 

RESULTS

Descriptive Statistics:

18. Include the results of a statistical comparison test of the variables between the different samples in table 2 and table 3 to assess their similarity of the samples

Answer: Addressed. A table comparing these samples is submitted as supporting material.

19. Given the highly bimodal nature of the observed utilities non-parametric summary measures (medians, IQR, etc.) and test statistics should be preferred added to the tables

Answer: Addressed. We report median EQ5D and IQR in the summary tables

Mapping results:

20. Show first the tables of performance and selection of the best fitting equations

then show the detailed equation of the best fitting one(s)

Answer: Addressed. Moved the equation after the table of performance. All other candidate models are shared in the supporting material.

21. Present the regression coefficients with their 95% CI and present also the variance-covariance matrix of the regression parameters

Answer: We have presented the regression coefficients for all candidate models (in manuscript text and supporting material).

DISCUSSION

22. Line 316 which generic PBM did Bremen used? Specify. How bad was the under-prediction of low observed utilities? Where was the utility threshold? How bad was the over-prediction for observed high utilities? Where was the utility threshold? What was the variance of the estimates compared to the observed variance of utilities for different values of utilities (low, average, high, perfect health) or per quartile?

Answer: Bremnan et al did not use a genetic PBM. Instead, they used a prostate cancer specific instrument, named PORPUS-U that measures health utilities. We did not report the other details around the performance of their algorithms as the instruments used in both studies are different. We merely wanted to bring to readers’ attention that this is the first study to map EPIC to obtain EQ5D utilities in our knowledge and that prior mapping studies in prostate cancer have used different instruments. The choice of a disease-specific PBM in the study by Bremnan et al. study makes the results harder to generalize across different therapeutic areas. Moreover, the algorithm has limited application as the most frequently employed PROM in clinical trials in prostate cancer is EPIC, while they mapped PCI, an older questionnaire that EPIC evolved from.

23. Line 338: Your risk of bias is linked to whether the censoring and non-response to the QoL questionnaires was truly random otherwise there is a risk of "survival or response" bias. Nothing tells you that the non responders had the same mapping coefficients as those of the completers so this could potentially alter the regression coefficients.

Answer: We investigated if differences exist between characteristics of responders and non-responders to gain insights into whether mapping coefficients between these patients would be different. Variables that differed between patients with missing and completed assessments: 

• Baseline: none

• 6 months: RT modality actually received (83.5% with completed EPICs received IMRT vs. 73.5% with missing EPICs)

• 12 months: age (60.6% with completed EPICs were >65 vs. 49.2% with missing EPICs) and planned RT modality (stratification factor; 81.9% with completed EPICs planned for IMRT vs. 75.2% with missing EPICs)

• 24 months: None were seen

• 60 months: race (83.7% with completed EPICs were white vs. 75.9% with missing EPICs) and ethnicity (98.6% with completed EPICs were not Hispanic vs. 94.6% with missing EPICs) and planned RT modality (stratification factor; 77.0% with completed EPICs planned for IMRT vs. 82.9% with missing EPICs)

As very few differences were seen between the responders and non-responders, the risk of response bias was considered to be low.

Reviewer #2: 

This paper has examined three econometric models for estimating EuroQol- 5 Dimension (EQ-5D) utility scores from the Expanded Prostate Cancer Index Composite (EPIC) to calculate quality adjusted life years for cost-utility analysis. The paper uses robust methods that should act as an aid for utility estimation within future economic evaluations of interventions using the Expanded Prostate Cancer Index Composite in Prostate Cancer. As such, it has the potential to act as a beneficial addition to the mapping literature. This article is well written, and the authors have carefully followed standard mapping methodology.

Response: We thank the reviewer for their insightful feedback. We have incorporated the feedback where applicable and provided clarification for the concerns raised in the review. The responses to each comment are summarized below.

Major comments:

1. Abstract Page 3, Line 51: The authors state that the lack of health utilities associated with the different health states assessed with the EPIC are unknown, therefore limiting the ability to perform cost-effectiveness evaluations. Can the authors edit this and use cost-utility analysis (CUA) and not cost-effectiveness analysis (CEA) as the form of economic evaluation which allows for the comparison of alternative treatment options in terms of incremental costs relative to quality-adjusted life-years (QALY) gained following treatment is a cost-utility analysis.

Answer: Thank you for the comment. We’ve addressed this and replaced ‘cost-effectiveness evaluations’ with ‘cost-utility analyses’ in the abstract.

2. Abstract Page 3, Line 52: The authors use the term "utility weights". This term is used in valuation studies when generating population preference weights or scoring algorithms and not mapping algorithms. The authors should correct this and use utility scores or utilities instead.

Answer: Thank you for the comment. We’ve addressed this as well.

3. Page 6: The authors present mapping as though the reader might already know what it is. Can the authors provide a more detailed definition of what mapping is?

Answer: The following sentence was added to provide more insights about the mapping process: “Utility mapping involves development and use of a statistical model or algorithm that links the outcomes from a PROM and a PBM to generate health utility values.”

4. Page 8 Line 162: Several other estimators have been applied in the mapping literature, including Fractional Logistic regression (FLOGIT); Censored Least Absolute Deviations (CLAD) regression; Generalized Additive Models; and finite mixture models. There are critics of the Tobit estimators, for example, but why haven't finite mixture models been applied?

Answer: In our study, the OLS models performed quite well with low errors overall both in the 5-fold cross-validation sample as well as in the 30% validation sample. Tobit and two-part models performed poorly compared to OLS models for each specification. These results were consistent with several other mapping studies that have found OLS models better than other more robust regression procedures in predicting health utilities. Therefore, additional model types were not explored.

5. Details of ethics committee approvals should be provided.

Answer: The following sentence was added to reflect the approvals sought in conducting this study: “The Institutional Review Board approval was sought and received from the University Of Maryland School Of Medicine and NRG Oncology”

6. Model selection should not be based solely upon the criteria, such as the predictive accuracy of on root mean square error (RMSE), laid out on page 12. The paper would be strengthened by a formal and staged selection process employed to choose between the models, including the BIC, AIC (for models for which the likelihood can be computed), misspecification tests, comparisons of conditional means or other similarly informative measures. These should dictate both the choice of covariates as well as the selection across different models.

Answer: We explored AIC and MAEs in addition to RMSE. MAEs do not penalize large errors like RMSE does, making RMSE a better indicator of predictive accuracy. The comparison of AICs for the tested model specifications provided results similar to the RMSEs, justifying the choice of RMSE as an indicator of predictive accuracy in our study.

7. Page 15: When assessing model performance: the errors should also be reported across subsets of the EQ5D utility score range as this is useful for indicating whether or not there is systematic bias in the predictions.

Answer: Based on the feedback from reviewer 1, we produced a Bland-Altman plot for the best performing model that shows the level of agreement between the observed and predicted utilities. Health utilities in our study wee under-predicted for patients in full health and over-predicted for those in more sever health states. This is a limitation of regression-based mapping and we have highlighted this in the discussion section. Additionally, we provide the plot of observed vs. predicted utilities, which show how accurate the prediction was (closer to the regression line indicating better prediction).

8. External validation is the preferred method for ascertaining the predictive accuracy of a mapping model. The authors of this paper use in-sample validation methods. Can the authors provide a detailed explanation of what a "five-fold cross-validation" is and how the in-sample validation datasets were generated? Secondly, how did they ensure that 'overfitting' was not an issue in the validation exercise? Thirdly, can the authors comment on how adequate five-fold cross-validation is as opposed to say ten-fold validation which has been in several mapping studies.

Answer: The following text is added in the manuscript to describe 5-fold-cross-validation: “In 5-fold cross-validation, the data are split into 5 equal parts and the model is fitted on 4 parts with the 5th being held out for validation. The fitted model of the 4 selected parts is used to compute the predicted residual sum of squares on the 5th omitted part, and this process is repeated for each of the 5 parts. The sum of the 5 predicted residual sums of squares is obtained for each fitted model and is the estimate of the prediction error. Indices such as the absolute mean of the residuals or errors (MAE), and square root of the mean of the residual sum of squares (RMSE) are used to determine model performance.” K-fold cross-validation ensures that we select the algorithm with the least errors on the training set as well as the test set, thus minimizing the risk of under-fitting or overfitting. The choice of k in k-fold cross-validation is somewhat arbitrary. While 10-fold CV has been found to result in models with relatively low bias and modest variance, 5-fold CV has also been used in several studies. On the other hand, some studies have used the leave one out cross-validation (LOOCV), where k = n. When selecting the number of folds in a CV exercise, one must balance the efficiency gains in terms of low bias, and the increase in run time and variance of the estimates as the number of folds increases. With a 5-fold CV approach, we had sufficient number of data points in our training sets (n ~ 400), increasing our confidence in this approach.

Minor comments:

1. Figure S1A: Please correctly label x-axis EQ5D and not EQ5D0

Answer: Addressed.

2. Page 8 Line 154: Please correct HRQOL to HRQoL

Answer: Addressed.

3. The paper does not seem to fully get across that mapping is a second-best solution and that having original data collected from relevant populations is a better solution. For the uninitiated, they may believe that EPIC data collected from patients with PC can be converted to EQ5D utilities "with a high level of accuracy". Hence, there is no need to collect original utility data.

Answer: The following sentence was added to the Introduction section to convey that mapping should be considered as an alternative only when direct estimation of utilities is not conducted: “Therefore, when utility information is not collected in a study, mapping has been proposed as an alternative solution and recommended as the second-best option after direct utility estimation for economic evaluations of interventions.”

---

## [Decision Letter · Decision Letter 1]

12 Mar 2021

Mapping Expanded Prostate Cancer Index Composite to EQ5D Utilities to Inform Economic Evaluations in Prostate Cancer: Secondary Analysis of NRG/RTOG 0415

PONE-D-20-34900R1

Dear Dr. Mishra,

We’re pleased to inform you that your manuscript has been judged scientifically suitable for publication and will be formally accepted for publication once it meets all outstanding technical requirements.

Kind regards,

A/Prof Michael E. O'Callaghan

Academic Editor

PLOS ONE

Additional Editor Comments (optional):

Reviewers' comments:

Reviewer's Responses to Questions

**Comments to the Author**

1. If the authors have adequately addressed your comments raised in a previous round of review and you feel that this manuscript is now acceptable for publication, you may indicate that here to bypass the “Comments to the Author” section, enter your conflict of interest statement in the “Confidential to Editor” section, and submit your "Accept" recommendation.

Reviewer #1: All comments have been addressed

Reviewer #2: All comments have been addressed

2. Is the manuscript technically sound, and do the data support the conclusions?

Reviewer #1: Yes

Reviewer #2: Yes

3. Has the statistical analysis been performed appropriately and rigorously? 

Reviewer #1: Yes

Reviewer #2: Yes

4. Have the authors made all data underlying the findings in their manuscript fully available?

Reviewer #1: No

Reviewer #2: No

5. Is the manuscript presented in an intelligible fashion and written in standard English?

Reviewer #1: Yes

Reviewer #2: Yes

6. Review Comments to the Author

Reviewer #1: (No Response)

Reviewer #2: The authors have addressed the points raised in my previous review, and publication of the manuscript as is recommended.

7. PLOS authors have the option to publish the peer review history of their article (what does this mean?). If published, this will include your full peer review and any attached files.

Reviewer #1: No

Reviewer #2: No

---

## [Editor Report · Acceptance letter]

5 Apr 2021

PONE-D-20-34900R1 

Mapping Expanded Prostate Cancer Index Composite to EQ5D utilities to inform economic evaluations in Prostate Cancer: Secondary analysis of NRG/RTOG 0415 

Dear Dr. Mishra:

I'm pleased to inform you that your manuscript has been deemed suitable for publication in PLOS ONE. Congratulations! Your manuscript is now with our production department. 

Kind regards, 

on behalf of

Associate Professor Michael E. O'Callaghan 

Academic Editor

PLOS ONE